# FrugalGPT: How to Use Large Language Models While Reducing Cost and Improving Performance

**Lingjiao Chen**                                                   *lingjiao@stanford.edu*
*Department of Computer Science*
*Stanford University*

**Matei Zaharia**                                                     *matei@berkeley.edu*
*Department of Electrical Engineering and Computer Sciences*
*University of California, Berkeley*

**James Zou**                                                         *jamesz@stanford.edu*
*Department of Biomedical Data Science*
*Stanford University*

**Reviewed on OpenReview:** *https://openreview.net/forum?id=cSimKw5p6R*

## Abstract

The rapid adoption of large language models (LLMs) has led to a growing number of companies offering generative LLMs as callable services at varying costs. We find that popular generative LLM APIs, such as GPT-4, Gemini 1.5, and Claude 3.5, exhibit heterogeneous pricing structures, with fees that can differ by two orders of magnitude and heterogeneous performance across tasks and input queries. This makes it challenging for users to decide which generative LLM APIs to utilize for their applications and budget. Motivated by these findings, we propose FrugalGPT, an algorithmic framework that adaptively selects which generative LLMs to use for different queries to reduce cost and improve accuracy. Our experiments demonstrate that, for a range of natural language tasks including news classification, reading comprehension, and scientific question answering, FrugalGPT can match the performance of the best individual generative LLM (e.g., GPT-4) with up to a 98% cost reduction or improve the accuracy over GPT-4 by 4% at the same cost. The ideas and findings presented in this paper lay a foundation for using LLMs sustainably and efficiently.

## 1 Introduction

We are currently witnessing a surge in the adoption of generative large language models (LLMs). The enticing potential of generative LLMs has led to a growing number of companies (such as AI21, Anthropic, Google, OpenAI, etc.) offering LLMs as callable services. Consequently, ML practitioners now frequently build applications by invoking these foundation models. For example, Tweet sentiment analysis is an official use case of ChatGPT (OpenAI, 2024a), Strabag uses Microsoft services to predict construction site risks (Microsoft, 2024), and Stripe uses GPT-4 to detect fraudulent behavior (OpenAI, 2024b).

However, practitioners often face challenges in deciding which generative LLM services to utilize for their applications and optimizing their budgets. The cost of generative LLM services can vary by up to two orders of magnitude: for instance, the prompt cost for 10M tokens is $300 for OpenAI's GPT-4 Turbo but only $1.5 for Google's Gemini 1.5 Flash 8B (as shown in Table 3). Smaller LLMs are generally more affordable, but their performance is comparatively limited (as depicted in Figure 1(d)). Larger LLMs like GPT-4 Turbo offer better performance but at the risk of escalating costs. In addition to their financial burden, employing larger LLMs incurs significant environmental and energy impacts (Bender et al., 2021; Wu et al., 2022; Schwartz et al., 2020).

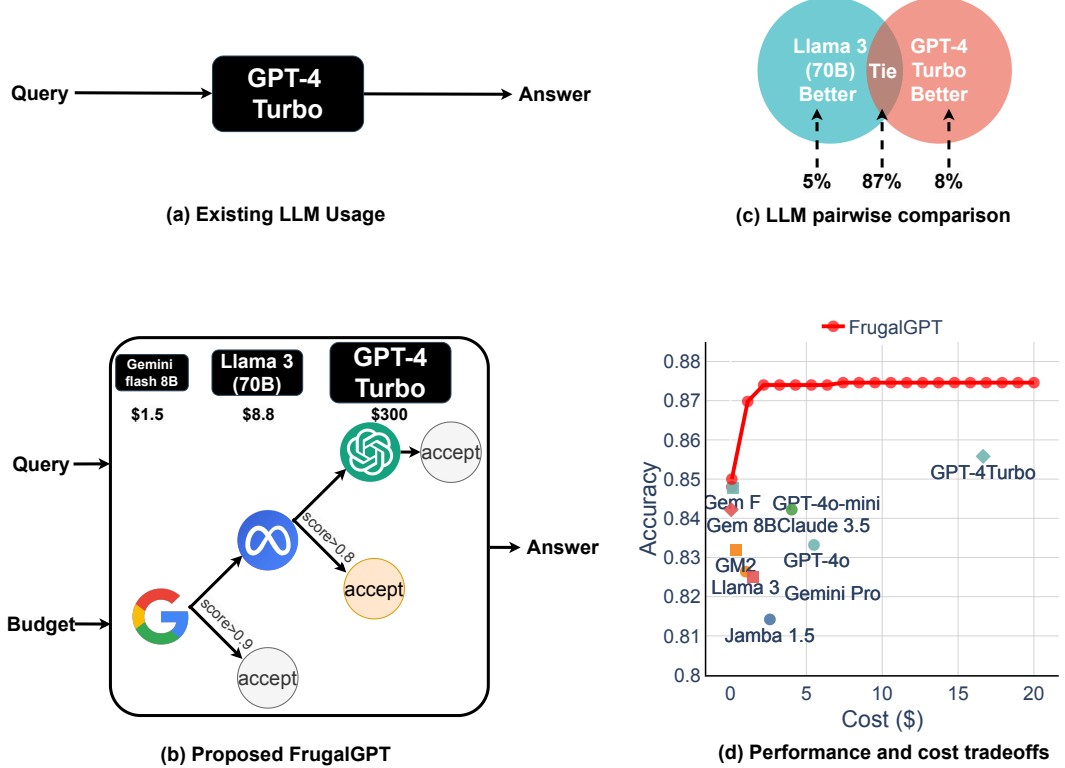

Figure 1: Comparisons of different approaches to using LLM services. (a) The standard usage sends queries to a single LLM (e.g., GPT-4 Turbo), which can be expensive. (b) FrugalGPT, adaptively decides which LLMs to trigger for different user queries to reduce the inference cost. By optimizing the selection of different LLM APIs (e.g., Gemini Flash, Llama 3 (70B), and GPT-4 Turbo), we can achieve substantial efficiency gains. (c) LLM performance breakdown on HEADLINES (a financial news dataset). Llama 3 (70B) outperforms GPT-4 Turbo on 5% of queries and produces identical generations on 87% of queries. (d) FrugalGPT can reduce the inference cost by 98% while exceeding the performance of the best individual LLM (GPT-4 Turbo) on HEADLINES. This is because FrugalGPT successfully learns data subsets on which inexpensive LLMs like Gemini Flash 8B are as good as or better than GPT-4 Turbo, and directs these data to the corresponding low-cost LLMs only.

In this paper, we empirically demonstrate that *for many of the tasks that generative LLMs are used for, it is possible to evaluate a result's quality using an inexpensive model.* For example, we found that DistillBERT can accurately predict the answer quality of many LLMs including GPT-4 Turbo and Llama 3 (70B) on common natural language tasks like classification and question answering. Furthermore, we find that *no generative LLM is "universally" superior to others.* Take the task of classifying price sentiments from news headlines as an example (Sinha & Khandait, 2021). There are 5% of queries where Llama 3 (70B) is entirely accurate while GPT-4 Turbo makes errors, and 87% of queries where both models provide identical responses (as illustrated in Figure 1(c)). Directing 92% (=87%+5%) of queries to Llama 3 (70B) and the remaining 8% to GPT-4 Turbo is considerably more cost-effective and performant than relying solely on GPT-4 Turbo. These discoveries suggest the possibility of *routing queries adaptively* to different LLMs to both lower the cost and enhance the performance of LLM applications.

Inspired by these findings, we propose FrugalGPT, an algorithmic framework that adaptively determines which LLMs to use given a user's budget. FrugalGPT first learns a *generation judger* that assigns a score to indicate the quality of different LLMs' generations for any given query. It then invokes a list of LLMs sequentially until the judger's score for an answer surpasses a threshold. For example, FrugalGPT may initially call Google Gemini 1.5 Flash 8B to obtain an answer. If the judger's score for this answer is lower than a threshold of 0.9, Llama 3 (70B) is subsequently invoked to generate a new response. The judger's

score for Llama 3's answer exceeds a threshold of 0.8, so no further LLMs are needed, and Llama 3's answer is returned to the user. Otherwise, GPT-4 Turbo is called to generate the final answer. We developed an efficient optimization technique to determine the optimal order of LLMs to call and the stopping threshold for each LLM as the core of FrugalGPT.

To demonstrate the potential of FrugalGPT, we implement and evaluate it on various tasks, such as news classification, reading comprehension, and scientific question answering, using real-world LLMs, including Claude 3.5 (Anthropic, 2024), Gemini 1.5 (Team et al., 2023), GPT-4o (Hurst et al., 2024), GPT-4 Turbo (OpenAI, 2023b), Jamba 1.5 Large (Team et al., 2024), and LLama 3 (70B) (Dubey et al., 2024). Our experiments show that FrugalGPT can save up to 98% of the inference cost of the best individual LLM API while matching its performance on the downstream task. On the other hand, FrugalGPT can improve performance by up to 4% at the same cost. This is because FrugalGPT accurately identifies queries on which some inexpensive LLMs are correct but the most powerful LLM (e.g., GPT-4) is incorrect, and directs these queries exclusively to the low-cost LLMs. We have also released the code and datasets used in our experiments at `https://github.com/stanford-futuredata/FrugalGPT`. We hope FrugalGPT paves the way for optimizing LLMs' inference cost and performance.

## 2 Related Works

**Model Ensembles.** Model ensembles (Dong et al., 2020; Ganaie et al., 2022), which involve combining multiple ML models for prediction, have gained popularity in supervised learning (García-Pedrajas, 2009; Friedman, 2002), unsupervised learning (Yang et al., 2014), semi-supervised learning (Gupta et al., 2022), and weakly supervised learning (Diba et al., 2017). Recent work (Arora et al., 2022) shows that fusing multiple generations from GPT-J (Wang & Komatsuzaki, 2021) can compete with GPT-3's performance, and synthesizing multiple open-source LLMs' generations leads to better performance than individual LLMs (Jiang et al., 2023). Model ensembles typically require white-box access to multiple models for training, but LLM APIs are often black-box. Moreover, model ensembles necessitate querying all models for any single query, thereby increasing costs.

**ML-as-a-Service and Cascades.** Generative LLM APIs constitute a crucial component of the rapidly expanding machine-learning-as-a-service (MLaaS) industry. Recent studies have demonstrated the diversity of different ML APIs' predictions (Buolamwini & Gebru, 2018; Koenecke et al., 2020; Chen et al., 2022b;a). The concept of using multiple services for speed is known as model cascade (Viola & Jones, 2004), which has been applied in predictive ML domains such as pedestrian detection (Cai et al., 2015), analytic systems (Kang et al., 2017) and facial recognition (Li et al., 2015; Sun et al., 2013). Recent work (Chen et al., 2020; 2022c) builds a customized cascade for cost reduction, with a focus on classification ML APIs. However, their approach needs to estimate the performance of an ML API without querying it, based on simple signals such as labels from a proxy model. Such pre-query estimation is challenging for generative LLM APIs, whose outputs encompass a much larger space. FrugalGPT overcomes this by creating a post-query quality estimator. Furthermore, for a given query, previous work invokes at most two APIs, while FrugalGPT allows invoking three or more given the vast number of LLM APIs. This renders it computationally more challenging to find the best calling strategies, and thus we also develop novel techniques to identify the optimal strategies efficiently (Section 4).

**Speculative Decoding.** Speculative decoding has recently emerged for LLM inference acceleration without retraining or model architecture modification (Leviathan et al., 2023; Chen et al., 2023; Sun et al., 2023; Spector & Re, 2023; Liu et al., 2023). Its goal is to provide the same output as a large LLM at lower latency. It relies on inexpensive LLMs for most generations and switches to costly LLMs when necessary. However, it requires access to the decoding module, which is not available for proprietary LLMs like GPT-4, and because it aims to give the same answer as the large LLM, it misses the opportunity to provide a better answer in cases where the small LLM is more accurate.

The remainder of the paper is organized as follows. We start by offering more context and the problem statement in Section 3. We present how FrugalGPT works in Section 4. Section 5 shows the empirical benefits of FrugalGPT using real-world LLM APIs (including GPT-3, ChatGPT, and GPT-4). We discuss future prospects in Section 6.

## 3 Scope and Problem Statement

**Natural language query answering.** In this paper, we concentrate on the standard natural language query answering task, where the objective is to answer a query $q$ sampled from a natural language query distribution $\mathcal{Q}$. Various real-world natural language tasks, such as news classification and commonsense reasoning, can be formulated as query-answering problems.

**LLM marketplace.** We consider answering queries via the LLM market, which comprises $K$ different LLM APIs, denoted by $\{f_i(\cdot)\}_{i=1}^K$. Each $f_i(\cdot) : \mathcal{P} \mapsto \mathcal{A}$ is a function that, given a prompt $p$ from the prompt space $\mathcal{P}$, generates an answer from the answer distribution $\mathcal{A}$. Note that to use LLM APIs, one has to convert each query $q$ to some corresponding prompt first. LLM APIs are associated with their own *cost*, typically consisting of three components: a portion proportional to the length of the prompt, a portion proportional to the length of the generated answer, and (sometimes) a fixed cost per query. Formally, given a prompt $p$, the cost of using the $i$th API is $c_i(p) \triangleq \tilde{c}_{i,2}\|f_i(p)\| + \tilde{c}_{i,1}\|p\| + \tilde{c}_{i,0}$, where $\tilde{c}_{i,j}, j = 0, 1, 2$ are constants.

**An illustrative example.** Adapting the case study provided by (Kaiser & Slowik, 2023), assume a small business operates a customer service using GPT-4. The company caters to 15,000 customers each month, with each customer asking three questions twice a week, totaling 360,000 queries per month. Suppose for each question, its prompt averages 1800 tokens and the answer is around 80 tokens (as estimated by (Kaiser & Slowik, 2023)). Considering that the input and response costs of GPT-4 are \$0.03 and \$0.06 per thousand tokens, the total monthly cost amounts to $360 \times (\$0.03 \times 1800 + \$0.06 \times 80) \approx \$21.2\text{K}$. Such a high cost is prohibitive for many small businesses.

**Problem statement: budget-aware LLM API usage.** Our primary goal in this paper is *leveraging LLM APIs within a budget constraint*. Formally, this can be formulated as maximizing the overall task performance $\mathbb{E}_{(q,a)\in\mathcal{Q}\times\mathcal{A}}[r(q, \hat{a}(s, q))]$, while ensuring the average cost is bounded by a user-defined value $b$, i.e., $\mathbb{E}_{(q,a)\in\mathcal{Q}\times\mathcal{A}}[c(s, q)] \leq b$. Here, $a$ denotes the correct answer to the query $q$, $\hat{a}(s, q)$ is the generated answer by some strategy $s$ for query $q$, and $c(s, q)$ is the associated cost for processing query $q$ using strategy $s$. The reward function $r(\cdot, \cdot)$ measures how closely the generated answer aligns with the user query.

## 4 FrugalGPT: A Cost-aware Paradigm to Leverage LLMs

This section presents FrugalGPT, a cost-aware approach designed to harness the power of multiple LLM services. We begin by outlining the FrugalGPT pipeline and explaining the functionality of each component. Subsequently, we delve into the construction of the FrugalGPT pipeline for any user budget.

**FrugalGPT Pipeline.** FrugalGPT comprises three main components: an LLM router, an answer scorer, and a stop judger. Given a user query $q$, the LLM router is first invoked to select an LLM to obtain its response to the query. Next, the generation scorer takes the query, the answer, and the selected LLM as input and generates a quality measurement as output. Based on the quality measurement and the invoked LLM service, the stop judger determines whether (i) to stop and return the answer, or (ii) to repeat the process of invoking the LLM router and generation scorer.

The LLM router consists of two parts. First, given the previously invoked LLM service $k'$, it selects the next LLM service to use, denoted by $k \triangleq \sigma(k')$, where $\sigma : \{\varnothing, 1, 2, \cdots, K\} \mapsto \{\varnothing, 1, 2, \cdots, K\}$ is a permutation of all LLM services (with $\varnothing$ representing no invocation). Second, it sends the query $q$ to the $k$th service and obtains the generation $f_k(q)$ as output. Although the service permutation could depend on the input query in principle, our instantiation adopts a query-agnostic permutation $\sigma(\cdot)$ for simplicity. As an example, consider the case of three models: GPT-4, GPT-Neo, and GPT-J. In this case $K = 3$. The LLM router may return GPT-J's generation for the first time it is invoked. For the second and third time, it gives output by GPT-Neo and by GPT-4, respectively. For the fourth time and beyond, it simply returns empty. This corresponds to the permutation $\sigma(GPT\text{-}J) = GPT\text{-}Neo, \sigma(GPT\text{-}Neo) = GPT\text{-}4, \sigma(GPT\text{-}4) = \varnothing$. Another instance is that the LLM router first invokes GPT-Neo, then GPT-J, and finally GPT-4. In this case, the permutation becomes $\sigma(GPT\text{-}Neo) = GPT\text{-}J, \sigma(GPT\text{-}J) = GPT\text{-}4, \sigma(GPT\text{-}4) = \varnothing$.

The generation scorer, denoted by $g_i(\cdot, \cdot) : \mathcal{Q} \times \mathcal{A} \mapsto [0, 1]$, generates a quality measurement given a query and an answer produced by the $i$th LLM API. Generally, the generation scorer can be any function such that its higher values strongly correlate with the input generation's quality. In our instantiation, we adopt a DistilBERT (Sanh et al., 2019) model tailored for regression as the generation scorer, as it is smaller, cheaper, and faster than LLM services while still providing a reliable quality measurement. Specifically, we have added a linear layer on top of the original DistilBERT that takes the last representation layer (768-dimensional) as input and produces a 2-dimensional output to encode the answer correctness. The maximum value of the last layer, normalized by softmax, is returned as the final score. We utilize the same embedding as DistilBERT, ensuring compatibility and seamless integration. For each LLM service, we have trained the model weights with (i) the query appended by the service's response as input features, and (ii) whether the response is correct as labels. We will present an ablation study on the generation scorer in Section 5.

The stop judger is responsible for deciding when to stop and return the answer to the user. As higher quality measurements indicate better generation quality, we use a threshold-based stop judger: return answer $a$ if the quality measurement $g_i(q, a)$ is higher than a threshold $\boldsymbol{\tau}_i$ and go back to the router otherwise. The threshold vector $\boldsymbol{\tau}$ controls the trade-offs between performance and cost: larger values often lead to better performance, while smaller values favor lower cost.

**Joint optimization of the FrugalGPT Pipeline.** Configuring the LLM router and stop judger appropriately is crucial to FrugalGPT. Technically, we need to configure (i) the LLM router's service permutation function $\sigma(\cdot)$ and (ii) the stop judger's threshold vector $\boldsymbol{\tau}$. Our goal is to maximize the expected reward on the query distribution while satisfying the user budget. This problem can be formally modeled as the following optimization problem:

$$
\begin{aligned}
\max_{\sigma(\cdot), \boldsymbol{\tau}} \quad & \mathbb{E}\left[r(q, f_z(q))\right] \\
s.t. \quad & \mathbb{E}\left[\sum_{z': z' = \sigma^{(t')}(\varnothing), t' \leq t} c_{z'}(q)\right] \leq b, \\
& t \in [L], z = \sigma^{(t)}(\varnothing), g_z(q, f_z(q)) > \boldsymbol{\tau}_z, \\
& \forall t' < t, z' = \sigma^{(t')}(\varnothing), g_{z'}(q, f_{z'}(q)) \leq \boldsymbol{\tau}_{z'}
\end{aligned}
\tag{1}
$$

Here, the objective is the expected performance (reward), the first constraint ensures the average cost is bounded by the budget, the second constraint indicates that the stop judge returns the answer at the $t$-th iteration, and the last constraint indicates that the LLM router and the generation scorer are called repeatedly for previous iterations. $L$ is a hyperparameter that controls the maximum number of LLM services to call for a query. Solving this problem is inherently challenging because the optimization space is vastly large. $\sigma(\cdot)$ is a permutation function over all possible LLM services, and exhaustive search takes $O(K^L)$ computations. Moreover, even if $\sigma(\cdot)$ is fixed, the problem is non-convex with respect to the threshold vector $\boldsymbol{\tau}$. In fact, we can show that even if the scorers are of high quality, the optimization problem is still NP-hard, formally stated as follows. We leave the proof in the appendix due to space constraints. Suppose the scorers are perfect, i.e., $g_i(q, a) > g_i(q, a') \Leftrightarrow r(q, a) > r(q, a')$ Then Problem (1) is an NP-hard problem.

To overcome this computational obstacle, we design a specialized optimizer for this problem. It (i) prunes the search space of $\boldsymbol{\sigma}(\cdot)$ by ignoring any consecutive selection of LLMs with small answer disagreement, and (ii) approximates the objective by interpolating it within a few samples.

Search space pruning removes candidate permutation functions with relatively small maximum performance improvement, or *MPI*. Here, *MPI* is a function of two LLMs, $k_1, k_2$, that measures at most how many mistakes $k_2$ incurs can be fixed by $k_1$. Formally, $MPI(k_1, k_2) \triangleq \Pr[r(q, f_{k_1}(q)) > r(q, f_{k_2}(q))]$. Suppose $k$ is called from the last iteration in the cascade. Then in the next iteration, calling any LLMs with small MPI would not yield significant performance gains and thus could be avoided. Inspired by this, we introduce the following pruning condition

Table 1: Summary of commercial LLM APIs. We use 14 LLM APIs from 6 providers. The cost was retrieved in March 2023. The cost includes three components: input (proportional to the number of input tokens), output (proportional to the number of generated tokens), and a fixed cost per request. LLMs' costs can differ by up to 2 orders of magnitude. For example, to process 1M input tokens, GPT-J from Textsynth costs only \$0.2, but OpenAI's GPT-4 costs \$30.

| Provider | API | Size/B | Cost (USD) | | |
| --- | --- | --- | --- | --- | --- |
| | | | 1M Input tok. | 1M Output tok. | Request |
| OpenAI | GPT-Curie | 6.7 | 2.00 | 2.00 | 0 |
| | ChatGPT | NA | 2.00 | 2.00 | 0 |
| | GPT-3 | 175 | 20.00 | 20.00 | 0 |
| | GPT-4 | NA | 30.00 | 60.00 | 0 |
| AI21 | J1-Large | 7.5 | 0.00 | 30.00 | 0.0003 |
| | J1-Grande | 17 | 0.00 | 80.00 | 0.0008 |
| | J1-Jumbo | 178 | 0.00 | 250.00 | 0.005 |
| Cohere | Xlarge | 52 | 10.00 | 10.00 | 0 |
| | Medium | 6.1 | 10.00 | 10.00 | 0 |
| Textsynth | GPT-J | 6 | 0.20 | 5.00 | 0 |
| | FAIRSEQ | 13 | 0.60 | 15.00 | 0 |
| | GPT-Neox | 20 | 1.40 | 35.00 | 0 |
| Databricks Model Serving | Dolly | 7 | 0.27 | 0.27 | 0 |
| ForeFrontAI | QA | 16 | 5.80 | 5.80 | 0 |

$$\sigma(k) \in \{k' \in K \mid MPI(k'', k) \geq MPI(k', k),$$
$$\text{for at most } m-1 \text{ other values of } k'' \in K\} \tag{2}$$

That is to say, given the previously invoked LLM $k$, the next LLM to call must hold the top-$m$ value of MPI with respect to $k$. This reduces the search complexity from $O(K^L)$ to $O(m^L)$. In practice, we found that $m = 3$ often suffices to identify a high-quality cascade.

Now suppose the function $\sigma(\cdot)$ is fixed. The remaining step is to find the optimal threshold vector $\boldsymbol{\tau}$. This can be resolved via a two-stage approximation. First, we divide the search space $[0, 1]^L$ into a few equal-size grids. Next, within each grid, we approximate the objective by a quadratic function of the threshold vector, whose parameters are determined by the grid bound values. Then within each grid, we can leverage a QP solver to find the optimal solution. The final solution is the best among all grids. The combination of the above two techniques provides an efficient implementation with satisfactory performance, as demonstrated later in Figure 3.

## 5 Experiments

In this section, we present an empirical study on FrugalGPT. Our goals are four-fold: (i) understand when and why FrugalGPT lowers the cost, (ii) quantify the cost savings attained by FrugalGPT while matching the best individual LLM API's performance, (iii) measure the trade-offs between performance and cost enabled by FrugalGPT, and (iv) explore how different factors including data distribution shifts and scorer's quality affect FrugalGPT.

**Setup: LLM APIs, Tasks, Datasets, and FrugalGPT instances.** We have selected 14 LLM APIs from 6 mainstream providers, namely, OpenAI (OpenAI, 2023a), AI21 (AI21, 2023), Cohere (Cohere, 2023), Textsynth (Textsynth, 2023), Databricks (Databricks, 2023), and ForeFrontAI (Forefront AI, 2023). The pricing information and all empirical results with these models were collected in March 2023, and results

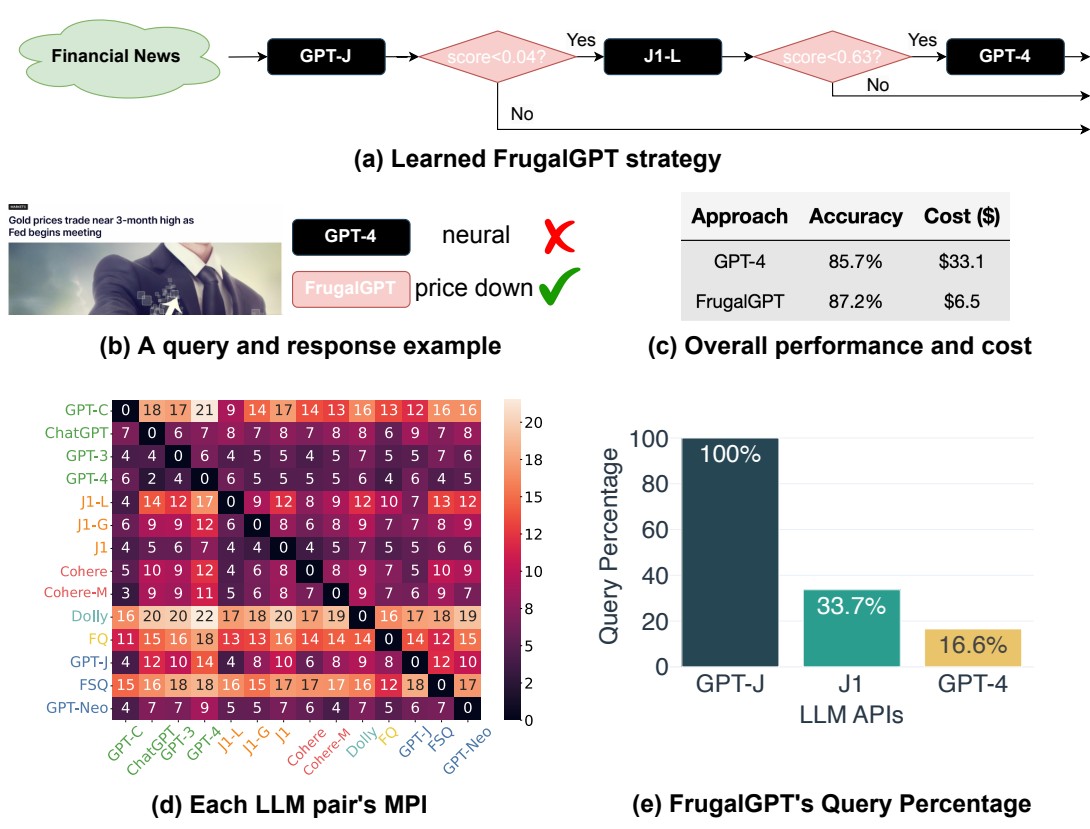

Figure 2: A case study of FrugalGPT on the HEADLINES dataset. (a) The learned FrugalGPT on this dataset with an overall budget of $6.5, one-fifth of GPT-4's cost. FrugalGPT avoids querying GPT-4 as long as GPT-J and J1-L produce high-quality answers. (b) Sometimes GPT-4 makes a mistake, but FrugalGPT learns to use the correct answers by J-1 and GPT-J. (c) Overall, FrugalGPT reduces the cost by 80%, while improving the accuracy by 1.5% compared to GPT-4. (d) The maximum possible improvement (MPI) for each LLM pair, measuring how often one LLM (each row) makes a mistake while another (each column) is correct. Even for the best individual LLM, GPT-4, cheap LLMs (e.g., GPT-J) can be better on 6% of the data. (e) FrugalGPT sends only 16.6% queries to GPT-4 and thus saves cost.

with newer models would be discussed later. The details are summarized in Table 1. FrugalGPT has been developed on top of these APIs and evaluated on a range of datasets belonging to different tasks, including HEADLINES (Sinha & Khandait, 2021), OVERRULING (Zheng et al., 2021), COQA (Reddy et al., 2019), AGNEWS (Zhang et al., 2015) and SCIQ (Welbl et al., 2017). More details of the datasets and tasks can be found in the Appendix. We focus on FrugalGPT with the hyperparameter $L = 3$, as this simplifies the optimization space and shows exciting results. Each dataset is randomly split into a training set (50%) to learn FrugalGPT and a test set for evaluation (50%).

**A Case Study.** We begin with a case study on the HEADLINES dataset. We set the budget to be $6.5, which is one-fifth of GPT-4's cost. As depicted in Figure 2 (a), the learned FrugalGPT sequentially calls GPT-J, J1-L, and GPT-4. For any given query, it first extracts an answer from GPT-J. If the score of this answer is greater than 0.96, the answer is accepted as the final response. Otherwise, J1-L is queried. J1-L's answer is accepted as the final response if its score is greater than 0.37; otherwise, GPT-4 is invoked to obtain the final answer. Interestingly, this approach outperforms GPT-4 for numerous queries. For instance, given a headline "Gold prices trade near 3-month high as Fed begins meeting" from NASDAQ, FrugalGPT accurately predicts that the price is going down, while GPT-4 provides an incorrect answer (as shown in Figure 2(b)). Overall, FrugalGPT results in both accuracy gains and cost reduction, as illustrated in Figure 2(c). This is partially because FrugalGPT only sends 16.6% queries to GPT-4 (see Figure 2(e)).

Table 2: Cost (USD) savings by FrugalGPT to match the best individual LLM's performance.

| Dataset | Type | Best single LLM | Cost to reach the same accuracy (USD) | | Cost Savings |
|---------|------|-----------------|----------------|-----------|--------------|
| | | | Best single LLM | FrugalGPT | |
| HEADLINES | 4-way classification | GPT-4 | 33.1 | 0.6 | 98.3% |
| OVERRULING | 2-way classification | GPT-4 | 9.7 | 2.6 | 73.3% |
| COQA | Free-form QA | GPT-3 | 72.5 | 29.6 | 59.2% |
| AGNEWS | 4-way classification | GPT-4 | 64.6 | 15.9 | 75.4% |
| SCIQ | Free-form QA | GPT-3 | 132.4 | 63.1 | 52.3% |

**LLM diversity.** Why can multiple LLM APIs potentially produce better performance than the best individual LLM? Similar to how ensemble methods can improve accuracy on many tasks, this is often due to generation diversity: even an inexpensive LLM can sometimes correctly answer queries on which a more expensive LLM fails. Recall that we introduce maximum performance improvement ( *MPI*) in Section 4 as an pruning metric. It also measures the generation diversity well: larger value of MPI indicates that one generative LLM give more responses different from another one. As shown in Figure 2 (d), MPI is indeed large for many pairs of generative LLMs. For instance, there are 6% queries where GPT-4 is incorrect but GPT-J (and GPT-C, J1-L, or Dolly) can give desired answers. This indicates the potential of combining multiple generative LLMs, and verifies why FrugalGPT offers cost reduction without performance drops.

**Cost Savings.** Subsequently, we examine if FrugalGPT can reduce costs while maintaining accuracy and, if so, by how much. Table 2 displays the overall cost savings of FrugalGPT, which range from 50% to 98%. This is feasible because FrugalGPT identifies the queries that can be accurately answered by smaller LLMs and, as a result, only invokes those cost-effective LLMs. Powerful but expensive LLMs, such as GPT-4, are utilized only for challenging queries detected by FrugalGPT.

**Performance and Cost Trade-offs.** Now, we investigate the trade-offs between performance and cost achieved by FrugalGPT, as illustrated in Figure 3. Here we focus on three datasets due to space limitations; more results on other datasets can be found in the Appendix.

Several interesting observations can be made. First, the cost ranking of different LLM APIs is not fixed. For instance, J1 is the second most expensive LLM on the HEADLINES dataset, while GPT-3 holds that position on the OVERRULING and COQA datasets. This is primarily due to the heterogeneous pricing mechanism: J1 incurs a high cost for each generated token but charges nothing for input tokens, whereas GPT-3 charges for both input and output tokens. Moreover, more expensive LLM APIs sometimes result in worse performance than their cheaper counterparts. For example, J1 is costlier than GPT-3 on HEADLINES, but its performance is inferior. These observations underscore the importance of aptly selecting LLM APIs, even in the absence of budget constraints. Next, we note that FrugalGPT enables smooth performance-cost trade-offs across all evaluated datasets. This offers flexible choices to LLM users and potentially helps LLM API providers save energy and reduce carbon emissions. In fact, FrugalGPT can simultaneously reduce costs and improve accuracy. For example, on the OVERRULING dataset, FrugalGPT achieves a 1% accuracy gain while reducing costs by 73% compared to the best LLM API, GPT-4. This is likely because FrugalGPT integrates knowledge from multiple LLMs.

The example queries shown in Figure 3 further aid in understanding why FrugalGPT can simultaneously improve performance and reduce costs. GPT-4 makes mistakes on some queries (e.g., the first example in part (a)), but some low-cost APIs provide correct predictions. FrugalGPT accurately identifies those queries and relies solely on the inexpensive APIs. For example, GPT-4 incorrectly infers no overruling from the legal statement "The time has come to reconcile and regularize our cases in this field," as shown in Figure 3(b). However, FrugalGPT accepts GPT-J's correct answer, avoiding the use of expensive LLMs and improving overall performance. Naturally, a single LLM API is not always correct; FrugalGPT overcomes this by employing a chain of LLM APIs. For example, in the second example shown in Figure 3(a), FrugalGPT

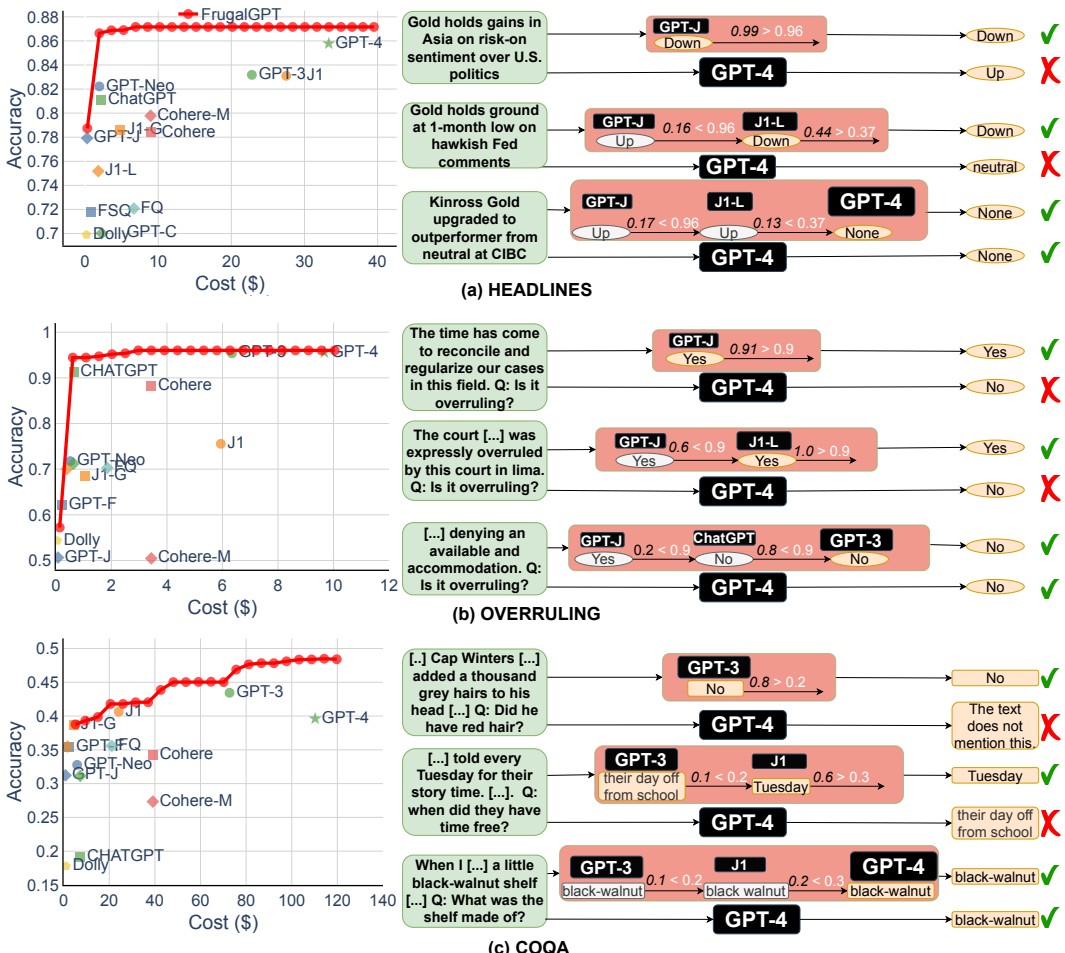

Figure 3: Accuracy and cost tradeoffs achieved by FrugalGPT. Overall, FrugalGPT often achieves the same performance of the best individual LLM API (e.g., GPT-4) with orders of magnitudes smaller cost. When incurring the same cost, FrugalGPT can improve the accuracy by up to 5%. Examples of FrugalGPT for each dataset are shown on the right. We show similar performance-cost tradeoff improvements for FrugalGPT for AGNEWS and SCIQ in the Appendix.

identifies that GPT-J's generation may not be reliable and turns to the second LLM in the chain, J1-L, to find the correct answer. Again, GPT-4 provides the wrong answer. FrugalGPT is not perfect, and there remains ample room for cost reduction. For example, in the third example in Figure 3(c), all LLM APIs in the chain give the same answer. However, FrugalGPT is unsure if the first LLMs are correct, resulting in the need to query all LLMs in the chain. How to avoid such cases is an interesting direction of future work.

**Comparison with a threshold baseline.** To further understand the benefits of FrugalGPT, we also compare it against a simple thresholding method. In particular, it first invokes GPT-J, and computes the log probability averaged over all output tokens. If the averaged log probability is larger than a threshold, GPT-J's response is accepted. Otherwise, GPT-4 is invoked. This can be seen as a variant of FrugalGPT with $L = 2$, except that the scorer is simply the first generator's own judgment. We evaluate this method with different threshold values on the HEADLINES dataset, and show its performance in Figure 4 as the blue dashed line (the blue texts correspond to the threshold values).

There are several interesting observations. First, this simple threshold method enables a good tradeoff between performance and cost. In fact, the corresponding blue in Figure 4 is much higher than a straight line between GPT-J and GPT-4. This suggests that the log probability returned by GPT-J indeed reflects

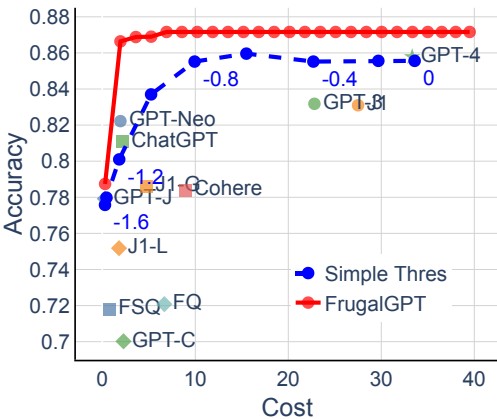

Figure 4: Comparison of FrugalGPT with a simple threshold baseline on the HEADLINES dataset. Overall, we observe that FrugalGPT consistently outperforms the threshold method given different cost constraints.

how likely the answer is correct. Second, a threshold value in the middle (-0.8) leads to the best performance. This again verifies our intuition that no single model is the best. Finally, FrugalGPT consistently outperforms this threshold method. For example, to match GPT-4's performance, this simple baseline reduces 60% cost while FrugalGPT reduces more than 90%. This is because FrugalGPT uses three models sequentially, and also uses an optimized generation scorer.

Table 3: Price comparison of models used for evaluation in 2024. The price was collected in November 2024. As of November 2024, Claude 3.5, GPT-4o, LLama 3 (70B), and Gamma 2 (9B) had different versions, and in our experiments we used `claude-3-5-sonnet-20240620`, `gpt-4o-2024-05-13`, `Meta-Llama-3-70B-Instruct-Turbo` and `gemma-2-9b-it` offered by the corresponding providers, respectively. Overall, the prices are lower than those in 2023, but there is still a large price difference between these new models. For example, the generation cost of GPT-4 Turbo is 200x of Google Flash 8B's cost.

| Provider | Model | Cost (USD / 1M tokens) | |
| --- | --- | --- | --- |
| | | Input | Output |
| AI21 | Jamba 1.5 Large | 2.00 | 8.00 |
| Anthropic | Claude 3.5 Sonnet | 3.00 | 15.00 |
| Google | Gemini 1.5 Pro | 1.25 | 5.00 |
| | Gemini 1.5 Flash | 0.075 | 0.30 |
| | Gemini 1.5 Flash 8B | 0.0375 | 0.15 |
| OpenAI | GPT-4o | 5.00 | 15.00 |
| | GPT-4 Turbo | 10.00 | 30.00 |
| | GPT-4o mini | 0.15 | 0.60 |
| Together AI | Llama 3 (70B) | 0.88 | 0.88 |
| | Gamma 2 (9B) | 0.30 | 0.30 |

**Evaluation using more recent models.** While the main evaluation was conducted using models available in 2023, many new models with lower prices have been developed since 2024. To understand how FrugalGPT works with these newly developed models, we evaluate its performance using a few more recent models on the HEADLINES and SCIQ datasets. In particular, we use eight proprietary models, GPT-4-Turbo, GPT-4o, and GPT-4o-mini offered by OpenAI, Jamba 1.5 Large from AI21, Claude 3.5 Sonnet from Anthropic, Gemini 1.5 Pro, Gemini 1.5 Flash, and Gemini 1.5 Flash 8B from Google, and two open-source models, LLama 3 (70B) and Gamma 2 (9B) provided by Together AI. This is because these models show superior

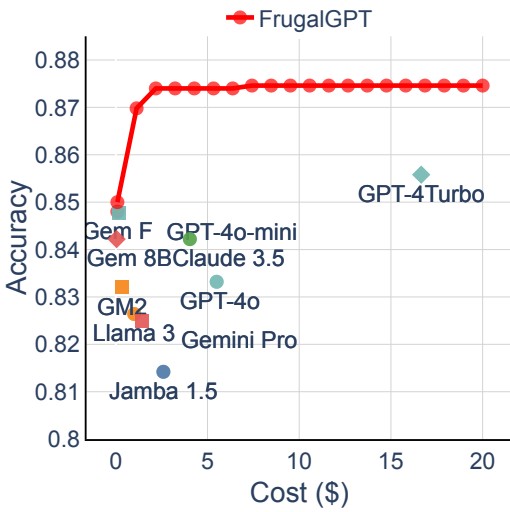 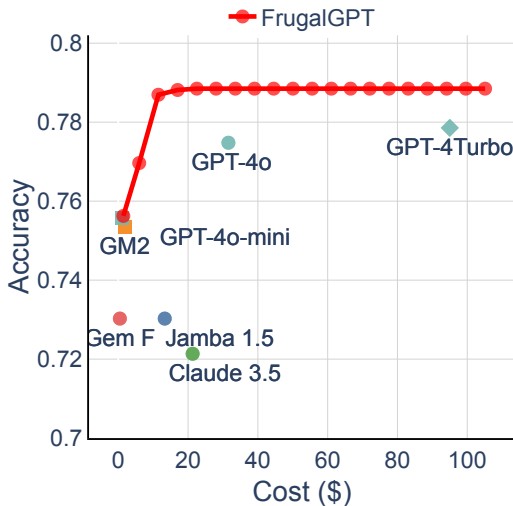

Figure 5: FrugalGPT's performance on the HEADLINES (left) and SCIQ (right) datasets using more recent models. In particular, we use eight proprietary models, GPT-4-Turbo, GPT-4o, and GPT-4o-mini offered by OpenAI, Jamba 1.5 Large from AI21, Claude 3.5 Sonnet from Anthropic, Gemini 1.5 Pro, Gemini 1.5 Flash, and Gemini 1.5 Flash 8B from Google, and two open-source models, LLama 3 (70B) and Gamma 2 (9B) provided by Together AI. We use the same prompts and temperatures for all these experiments. On SCIQ, we omit three models since their accuracy is much lower than other models, namely, LLama 3 70B (24.1%), Gemimi Pro (53.9%) and Gemini Flash 8B (43.6%). FrugalGPT can reduce the cost by more than 90% while matching the best individual model's performance on these datasets.

performance and are widely used in many applications. Their prices are summarized in Table 3, collected in November 2024. Compared to Table 1, the overall prices are indeed lower, but there is still a large price difference between these models.

Figure 5 shows the performance-cost tradeoff achieved by FrugalGPT using these new models on HEADLINES (left) and SCIQ (right). Overall, FrugalGPT consistently offers large cost savings. For example, to match GPT-4-Turbo's performance, it requires less than 10% of the cost. It is also interesting to note that expensive models again are not necessarily better than cheap models. For example, GPT-4o's performance is worse than GPT-4o-mini's on this task, although the former's price is much higher. This further underscores the importance of choosing models carefully even if the budget is large.

**Performance Resilience to Data Distribution Shifts.** A common challenge when deploying ML systems in practice is data distribution shifts, i.e., the queries encountered during deployment differ from those in development. To understand the robustness of FrugalGPT against data distribution shifts, we trained FrugalGPT on the original HEADLINES training data and evaluated its performance on four testing datasets with different distributions. Specifically, we created these testing datasets by altering the distribution of labels. For instance, in Variant 1, the label distribution is 33% (up), 17% (down), 17% (none), and 33% (neutral). Conversely, the original dataset's label distribution is balanced (25% for each label). Details can be found in Table 5 in the Appendix. As depicted in Figure 7(a) in the appendix, the performance of both FrugalGPT and GPT-4 remains relatively consistent across different data distributions. Interestingly, while using only 10% of GPT-4's cost, FrugalGPT often delivers similar or superior performance compared to GPT-4 under several testing data distributions.

**Effects of Scorer Functions.** The scorer plays a crucial role in FrugalGPT. Therefore, it is essential to study how the scorer's quality impacts FrugalGPT's performance. In this regard, we focused on three backbones for the scorer with varying numbers of parameters: ALBERT (11M), DistilBERT (67M), and BERT (110M). We trained the scorer on the HEADLINES dataset using different backbone models and compared the performance of the resulting FrugalGPT, with a budget of 10% of GPT-4. As illustrated

in Figure 7(b), a low-quality scorer (such as ALBERT) indeed leads to limited performance, as expected. Conversely, larger scorers with better quality, such as DistilBERT and BERT, offer higher performance.

## 6 Discussions and Future Prospects

The substantial cost of employing LLMs in real-world scenarios presents a considerable barrier to their widespread usage. In this paper, we discovered that for many tasks on which LLMs are commonly used today, (i) small models can predict the quality of LLMs accurately, and (ii) no LLM is universally better than others. Based on these findings, we introduce FrugalGPT, our approach to resolving the cost challenge. Our empirical findings show that FrugalGPT can reduce costs by up to 98% while preserving the performance of cutting-edge LLMs.

There are many interesting directions for future exploration. For example, while FrugalGPT concentrates on balancing performance and cost, real-world applications call for the evaluation of other critical factors, including latency, fairness, and environmental impact. Incorporating these elements into optimization methodologies while maintaining performance and cost-effectiveness is an important avenue for future research. Furthermore, utilizing LLMs in risk-critical applications necessitates the careful quantification of uncertainty in LLM-generated outputs. We have released our code and datasets at https://github.com/stanford-futuredata/FrugalGPT, hoping to stimulate further research on generative service utilization.

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
