## Acknowledgement

This work was supported in part by a Google PhD Fellowship, a Sloan Fellowship, NSF CCF 1763191, NSF CAREER AWARD 1651570 and 1942926, NIH P30AG059307, NIH U01MH098953, grants from the Chan-Zuckerberg Initiative, Sutherland, and affiliate members and other supporters of the Stanford DAWN project and UC Berkeley SKY Lab, including Google, IBM, Intel, Lacework, Meta, Microsoft, Mohamed Bin Zayed University of Artificial Intelligence, Nexla, Samsung SDS, Uber, and VMware. We also thank anonymous reviewers for helpful discussion and feedback.

## A    Discussions on Other Strategies

Now we present our vision on two other strategies to use LLM APIs within a budget, namely, *prompt adaptation*, and *LLM approximation.* We give a few examples in Figure 6.

**Strategy 1: Prompt adaptation.**    The cost of an LLM query increases linearly with the size of the prompt. Consequently, a logical approach to reduce the cost of using LLM APIs involves decreasing the prompt's size, a process we refer to as prompt adaptation. *Prompt selection* (as illustrated in Figure 6 (a)) is a natural example of prompt adaptation: rather than employing a prompt containing numerous examples that demonstrate how to perform a task, one can retain a small subset of examples in the prompt. This results in a smaller prompt and subsequently lower cost. An intriguing challenge of prompt selection lies in determining which examples to maintain for various queries without compromising task performance.

An additional instantiation is *query concatenation* (Figure 6 (b)). It is important to note that processing queries individually necessitates sending the same prompt to an LLM API multiple times. Therefore, the fundamental concept of query concatenation involves sending the prompt only once to the LLM API while allowing it to address multiple queries, thereby preventing redundant prompt processing. To accomplish this, several queries must be concatenated into a single query, and the prompt must explicitly request the LLM API to process multiple queries. For instance, to handle two queries using one prompt, the examples presented in the prompt can include both queries followed by their corresponding answers.

**Strategy 2: LLM approximation.**    The concept of *LLM approximation* is quite simple: if an LLM API is too costly to utilize, one can approximate it using more affordable models or infrastructures. One example is the *completion cache*: as depicted in Figure 6 (c), the fundamental idea involves storing the response locally in a cache (e.g., a database) when submitting a query to an LLM API. To process a new query, we first verify if a similar query has been previously answered. If so, the response is retrieved from the cache. An LLM API is invoked only if no similar query is discovered in the cache. The completion cache provides substantial cost savings when similar queries are frequently posed. For instance, consider a search engine powered by an LLM API. If numerous users search for the same or similar keywords simultaneously, the completion cache facilitates answering all their queries by invoking the LLM only once.

Another example of LLM approximation is *model fine-tuning.* As shown in Figure 6(d), this process consists of three steps: first, collect a powerful but expensive LLM API's responses to a few queries; second, use the responses to fine-tune a smaller and more affordable AI model; and finally, employ the fine-tuned model for new queries. In addition to cost savings, the fine-tuned model often does not require lengthy prompts, thus providing latency improvements as a byproduct.

**Compositions.**    Combining approaches within and across different strategies can lead to further cost reduction and performance enhancement. For instance, *joint prompt and LLM selection* is a composition of prompt selection and LLM cascade: for a given query, it searches for the smallest prompt and most affordable LLM that achieves satisfactory task performance. Another example is to search across both existing LLM APIs and fine-tuned models. Note that the composition of different approaches also increases the computational costs for training. Consequently, this paves the way for investigating trade-offs between query costs, task performance, and computational costs.

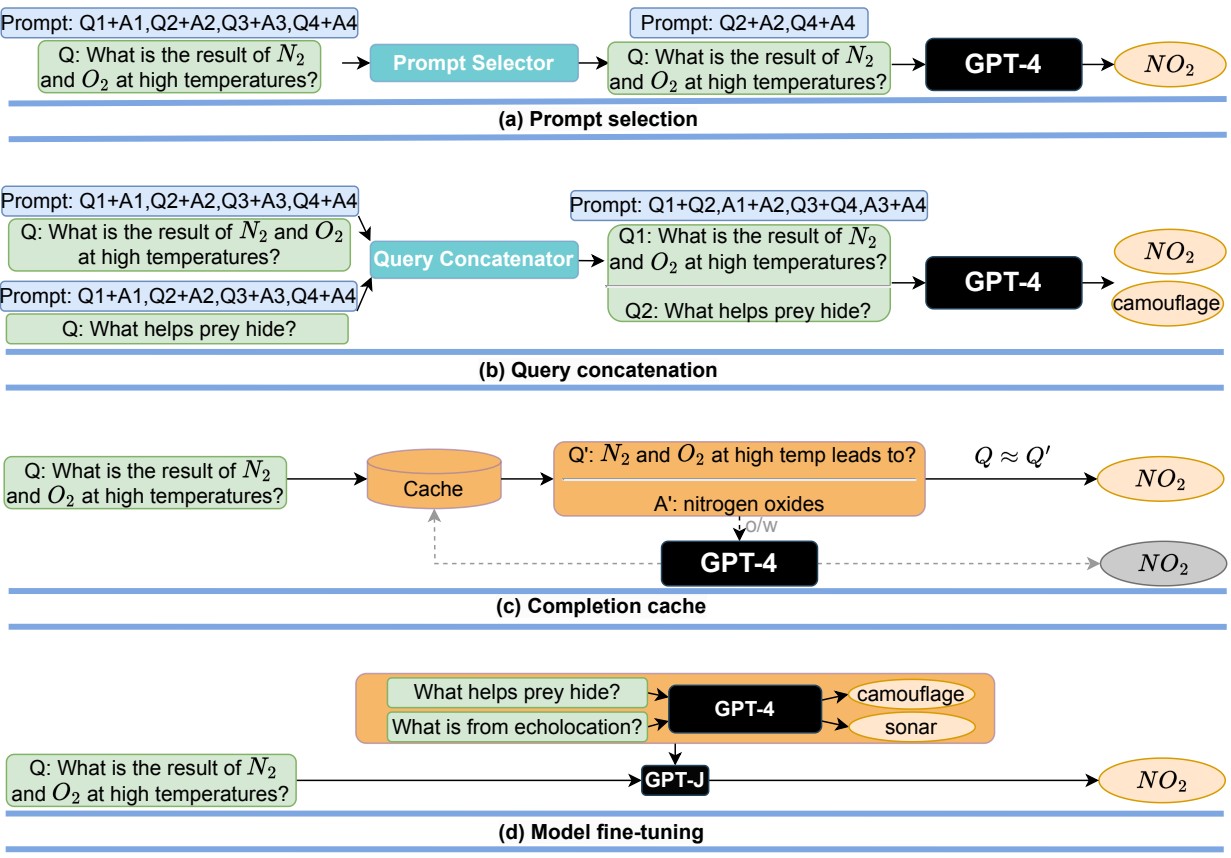

Figure 6: Illustrations of additional cost-saving strategies. (a) Prompt selection uses a subset of in-context examples as the prompt to reduce the size of the prompt. (b) Query concatenation aggregates multiple queries to share prompts. (c) Completion cache stores and reuses an LLM API's response when a similar query is asked. (d) Model fine-tuning uses expensive LLMs' responses to fine-tune cheap LLMs.

# B    Missing Proofs

Here we restate the Theorem 4 and then give its proof.    Suppose the scorers are perfect, i.e., $g_i(q, a) > g_i(q, a') \Leftrightarrow r(q, a) > r(q, a')$ Then Problem (1) is an NP-hard problem.

*Proof.* We can prove this by a reduction from the maximum coverage problem. Suppose that Problem (1) can be solved in polynomial time. For sufficiently sufficiently large $b$, it must also be solved in polynomial time. That is to say, the following problem can be solved in polynomial time.

$$\max_{\sigma(\cdot), \boldsymbol{\tau}} \ \mathbb{E}\left[r(a, f_z(q))\right]$$
$$s.t. t \in [L], z = \sigma^{(t)}(\varnothing), g_z(q, f_z(q)) > \boldsymbol{\tau}_z,$$
$$\forall t' < t, z' = \sigma^{(t')}(\varnothing), g_{z'}(q, f_{z'}(q)) \leq \boldsymbol{\tau}_{z'}$$

(3)

Now let us consider the maximum coverage problem

$$\max_I |\cup_{i \in I} S_i|$$
$$s.t. \ I \subseteq [|S|], |I| \leq L$$

(4)

where $S = \{S_1, S_2, \cdots, S_K\}$ is a collection of sets. Our main observation is that we can use the solver to Problem (3) as a subroutine to identify the solution to Problem (4). To see this, construct a uniform data

distribution whose support is all elements in $|\cup_{S_i \in S} S_i|$. Now, construct $K$ ML services $f_i(\cdot)$, such that

$$r(q, f_i(q)) = \begin{cases} 1 & \text{if } q \in S_i, \\ 0 & \text{otherwise.} \end{cases}$$

That is, the $i$th service is correct on a query if and only if the query is contained in the set $S_i$. Now let us use the polynomial solver to solve the Problem (3), which gives the optimal solution $\sigma^*(\cdot), \tau^*$. Now, the index $I = \{\sigma^{*(t)}(\emptyset)\}_{t=1}^{L}$ is the optimal solution to Problem (4). This is true for two reasons. First, note that the optimal value of Problem (4) is an upper bound of the optimal value of Problem (3). This is easy to see, as the reward value $r(f_z(q))$ is correct implies at least one ML service in the first $L$ service is correct. Second, this upper bound is achievable. To see this, let us set up the thresholds $\tau = 0.5$. Since the scorers are perfect, the final response is correct if and only if at least one ML API chosen by $\sigma()$ in the first $L$ step is correct. Thus, solving Problem (3) in polynomial time indicates solving Problem (4) in polynomial time. However, the Problem (4) is known to be NP-hard. Therefore, we can conclude that Problem (3) is NP-hard and thus the original Problem (1) is NP-hard.

□

## C  Experiment Setups and Extra Results

We provide more details on the experiment setups and extra empirical results here.

### C.1  Tasks and Datasets

Table 4: Summary of datasets used in the FrugalGPT LLM cascade experiments.

| Dataset | Domain | Size | #Examples in the prompt |
|---------|--------|------|-------------------------|
| HEADLINES | Finance | 10000 | 8 |
| OVERRULING | Law | 2400 | 5 |
| COQA | Multi-domain | 7982 | 2 |
| AGNEWS | Journalism | 7600 | 8 |
| SCIQ | Science | 11680 | 8 |

We have evaluated FrugalGPT on five different datasets, ranging from domain-specific classification tasks to general-purpose question answering. The details are summarized in Table 4.Specifically, HEADLINES (Sinha & Khandait, 2021) is a financial news dataset where the goal is to determine the gold price trend (up, down, neutral, or none) by reading financial news titles. This is especially useful for filtering relevant news in financial markets. OVERRULING (Zheng et al., 2021) is a legal document dataset where the goal is to determine whether a given sentence is an overruling, i.e., rejecting previous legal cases. COQA (Reddy et al., 2019) is a multi-domain reading comprehension dataset developed in a conversational setting, which we have adapted as a direct query-answering task. AGNEWS (Zhang et al., 2015) is a news dataset. The task is to classify each news into one of four categories (business, world, sports, and sci/tech). SCIQ (Welbl et al., 2017) is a scientific question-answering dataset. Given a short paragraph, the target is to answer a question about physics, chemistry, and biology.

We reformat all datasets so that the LLMs process each data point as a few-shot learning problem. In particular, we append (i) a short description of the task alone with (ii) a few examples in the prompt to each data point and then feed it to the LLMs. The details of the prompt prefix is given in the following subsection. For AGNEWS, we randomly select the examples from their original training partitions. There are no official train-eval partitions for HEADLINES and OVERRULING, so we randomly select a few in-context examples from the entire datasets, and evaluate the performance of commercial APIs and FrugalGPT on the remaining

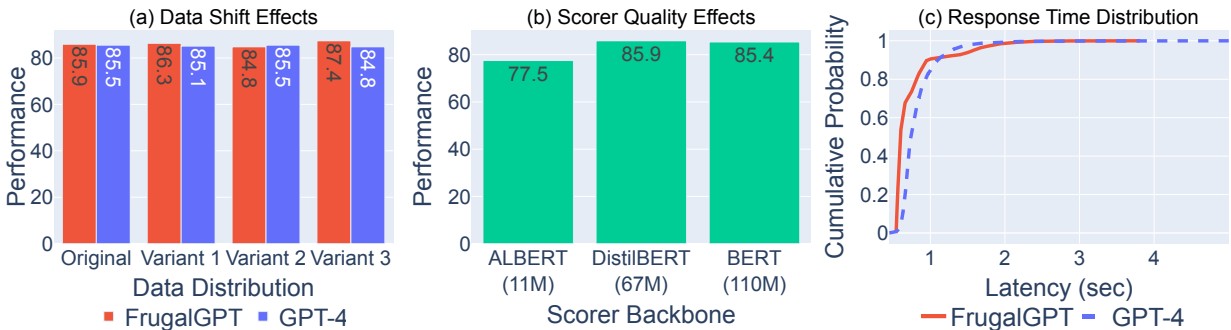

Figure 7: Ablation study of FrugalGPT with a budget of 10% of GPT-4 on the HEADLINES dataset. (a) Effects of data distribution shifts. Each variant corresponds to one label shift instance. (b) Effects of scorer quality. (c) Latency (response time) distribution. Overall, the performance of FrugalGPT remains relatively consistent under various testing data distributions different from its training data. As expected, a small and low-quality scorer, such as ALBERT, leads to limited performance, while larger and higher-quality scorers (DistilBERT and BERT) yield better performance. FrugalGPT calls cheaper and faster LLMs on most queries, resulting in shorter response times than GPT-4.

data points. AGNEWS, COQA, and SCIQ all offer their official training and evaluation partitions. Due to budget limits, in-context examples are randomly selected from the larger partitions (often the training partition) and the evaluation is performed on the evaluation partitions only. The training partition of SCIQ is relatively smaller so we evaluate the performance on the larger partition and select the in-context examples from the smaller partition.

## C.2 Additional Evaluations

Here we present additional empirical evaluation of FrugalGPT. In particular, we offer (i) the missing figure for an ablation study on HEADLINES, (ii) LLM diversity measurement, (iii) performance-cost tradeoffs on more datasets, (iv) data shift synthesis process, (v) comparisons with ensemble approaches, and (vi) effects of the training dataset.

**The Missing Figure for the Ablation Study of FrugalGPT on HEADLINES.** The missing figure for the ablation study on HEADLINES is given in Figure 7. As analyzed in the main paper, FrugalGPT exhibits robustness to distribution shifts. In addition, the scorer backbone model's quality does play an important role in FrugalGPT's performance. First, we observe that scoring/judging is easier than generating. E.g., a small model (DistillBERT with 67M parameters) is sufficient to accurately score the quality of large models' answers on the HEADLINES datasets. Second, we observe that the scorer's power, approximated by its size, has a significant effect on the overall performance. Indeed, increasing the size of the scorer from 11M (ALBERT) to 67M (DistillBERT) brings an 8% performance gain. Finally, this effect is not necessarily linear. In fact, further increasing the scorer's size from 67M to 110M results in a similar performance. Overall, this suggests that higher accuracy and lower performance can be achieved simultaneously by using a small and cheaper scorer, although the scorer must be carefully evaluated and chosen.

**Improved Latency.** The increasing size of LLMs often correlates with better performance but at the expense of longer response times. In addition to the main paper's analysis, we compare the response times of FrugalGPT and GPT-4. Specifically, we set the budget of FrugalGPT to be 10% of GPT-4's cost and compared their performance on the HEADLINES dataset. Overall, we observe that FrugalGPT is often much faster than GPT-4. For instance, 90% of the queries can be answered within 0.9 seconds by FrugalGPT, but more than 1.1 seconds by GPT-4, as shown in Figure 7(c). This is because FrugalGPT learns to call cheaper and faster LLMs for many queries, only invoking the expensive and slow GPT-4 when necessary. Although not explicitly optimized for latency, FrugalGPT inherently provides shorter response times for most queries and is thus desired for latency-critical applications.

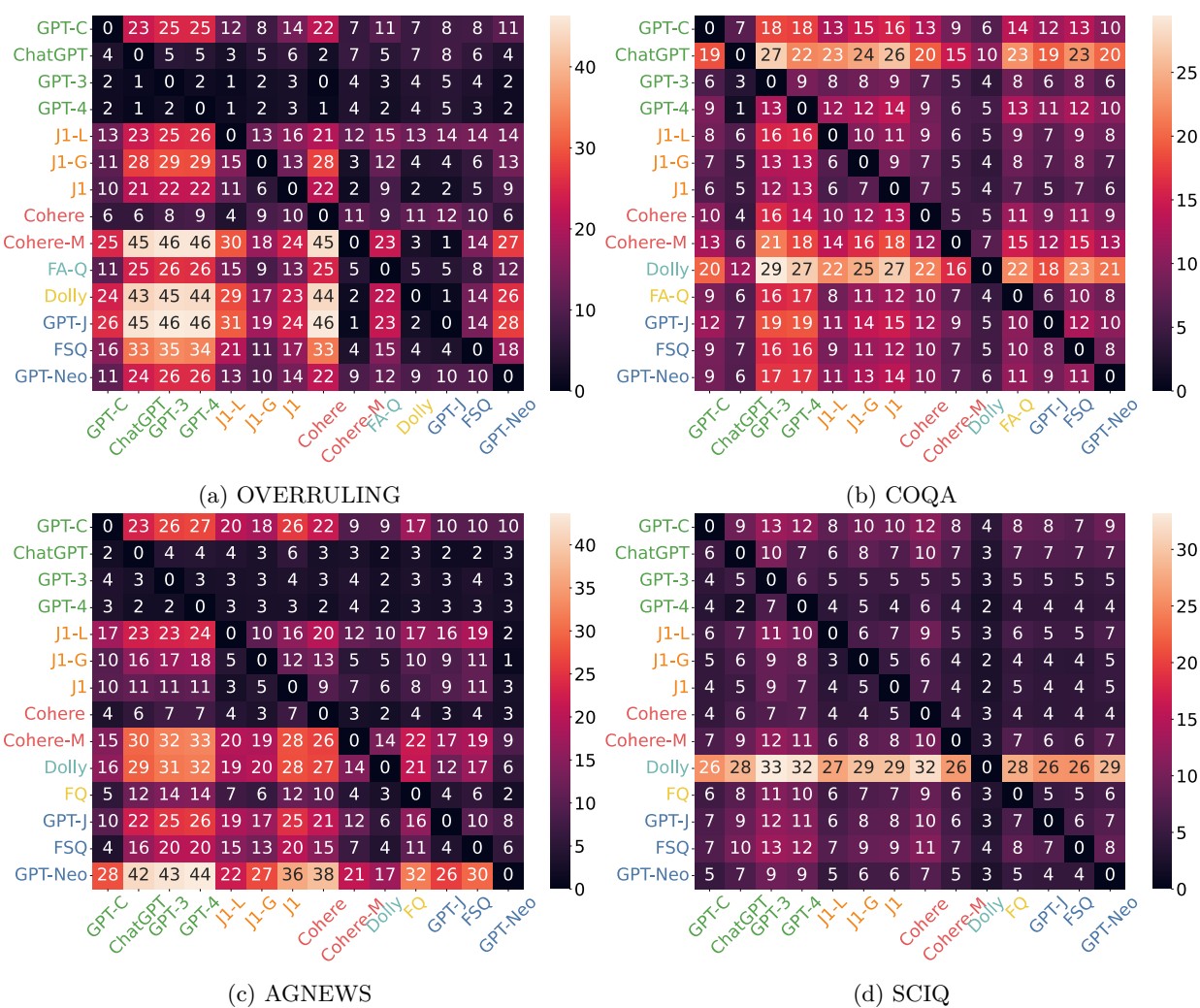

Figure 8: Maximum performance improvement (MPI) of each pair of LLMs. (a), (b), (c) and (d) correspond to the four datasets, separately. One entry indicates the percent of cases that the LLM on its row is wrong but the LLM on its column gives the right answer. Overall, we observe that cheap LLMs can be complementary to expensive ones quite often. For example, for about 5% of the data, GPT-4 makes a mistake but GPJ-J gives the right answer on OVERRULING.

**LLM Diversity.** In Section 5, the study of the MPI for HEADLINES reveals a large potential for performance improvements over the best individual LLM. Does this generalize to other datasets? Here, MPI between each pair of LLM APIs for the remaining 4 datasets is displayed in Figure 8. Overall, we observe a similar phenomenon. For instance, GPT-J, can enhance GPT-4's performance by up to 5% on the OVERRULING dataset. On the COQA dataset, there are 13% of data points where GPT-4 makes an error, but GPT-3 provides the correct answer. Although these improvements' upper bounds may not always be attainable, they demonstrate the possibility of utilizing more affordable services to achieve better performance. How to reach these upper bounds is an interesting open problem.

**Additional Performance-Cost Trade-offs.** The performance and cost trade-offs attained by FrugalGPT on AGNEWS and SCIQ are presented in Figure 9. The overall trends are the same as the other datasets shown in Figure 3: FrugalGPT offers flexible budget-aware strategies as well as significant cost reduction when matching the performance. The examples (on the right panel of the figure) provide a few dataset-specific insights. For example, GPT-4 incorrectly classified the first example in AGNEWS as "Worlds"

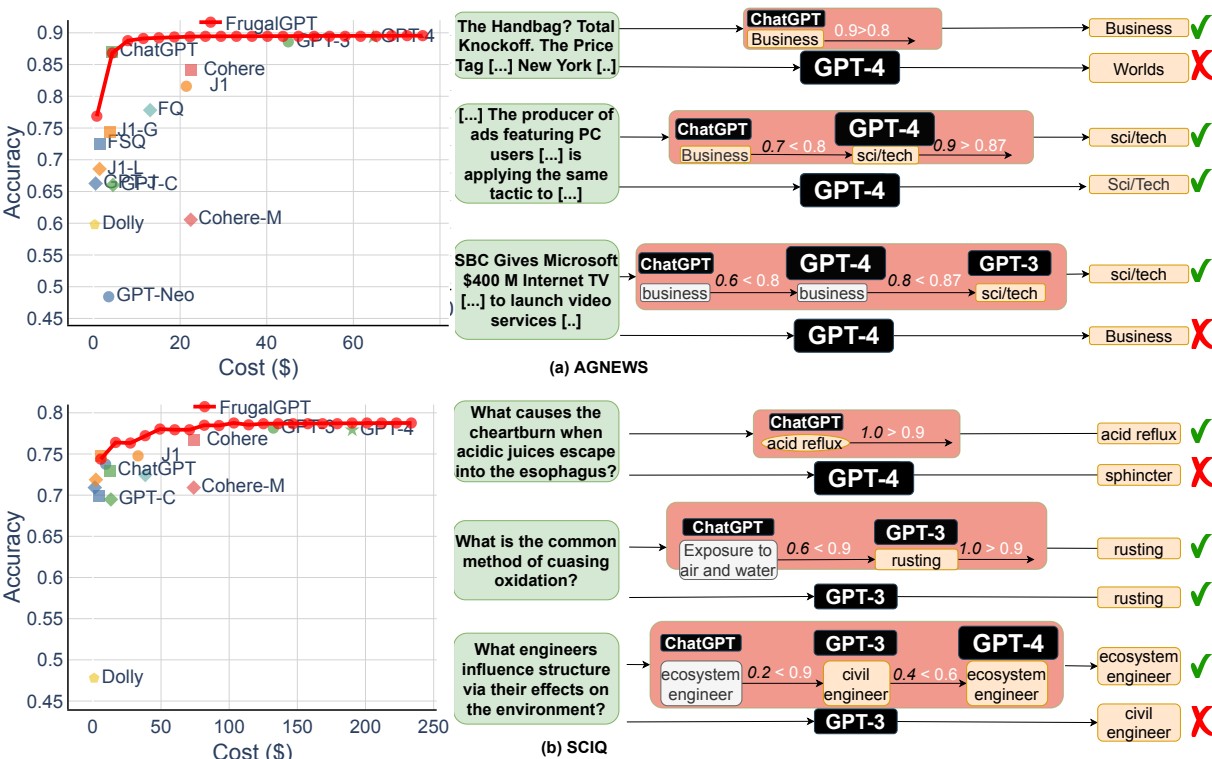

Figure 9: Accuracy and cost trade-offs achieved by FrugalGPT on AGNEWS and SCIQ. Overall, FrugalGPT achieves the same performance of the best individual LLM API (e.g., GPT-4) with orders of magnitudes smaller cost. Examples of LLM cascade for each dataset are shown on the right.

Table 5: Synthesized data distribution shifts for the ablation study. The label distribution of the original evaluation data is balanced. The three variants are all generated by varying the marginal label distribution. The table shows the fraction of different labels for each variant.

| Eval Data Instance | Label Distribution | | | |
|---|---|---|---|---|
| | Up | Down | None | Neutral |
| Original | 0.25 | 0.25 | 0.25 | 0.25 |
| Variant 1 | 0.33 | 0.17 | 0.17 | 0.33 |
| Variant 2 | 0.40 | 0.20 | 0.20 | 0.20 |
| Variant 3 | 0.50 | 0.17 | 0.17 | 0.17 |

while the correct answer is "business" as produced by FrugalGPT. This is probably because there are a few city names (e.g., New York) in the original news. The first example in the SCIQ dataset is also quite interesting. GPT-4's answer, "sphincter", was incorrect but reasonable: sphincter does offer the power for many movements in the stomach. However, it is not as precise as "acid reflux", the correct answer produced by FrugalGPT. This suggests that GPT-4 may have difficulty distinguishing high-level concepts from low-level concepts within certain contexts. FrugalGPT, on the other hand, learns to identify this and adapt to the most trustful LLM as needed.

Table 6: Comparison of FrugalGPT with ChatGPT and ChatGPT ensemble on the HEADLINES dataset. The ChatGPT ensemble approach queries ChatGPT three times with different temperatures (0, 0.1, 0.2) and then takes the majority vote as the final output. Overall, FrugalGPT achieves better performance than ChatGPT ensemble with a lower cost.

| Approach | ChatGPT (1 output) | ChatGPT ensemble (3 outputs) | FrugalGPT |
|---|---|---|---|
| Accuracy | 81.0% | 83.8% | 87% |
| Cost | $2.22 | $6.66 | $6.5 |

Table 7: Training size effects of FrugalGPT on the HEADLINES dataset. Overall, we notice that one thousand samples are sufficient to learn a FrugalGPT strategy with high-quality.

| Training set size | Accuracy (%) |
|---|---|
| 500 | 85.9 |
| 1000 | 86.8 |
| 2000 | 86.8 |
| 3000 | 86.8 |
| 4000 | 87.2 |
| 5000 | 87 |

## C.3 Data Shift Synthesis

To study the robustness to data distribution shifts, we have synthesized three data distributions different from the original data distribution (Figure 7). Note that the original data distribution can be decomposed as $\Pr[x, y] = \Pr[y] \Pr[x|y]$. To generate the variants, we fixed $\Pr[y|x]$ and varied the marginal label distribution. The corresponding label distribution is shown in Table 5. Here, we observe a relatively large distribution shift. For example, the number of news showing increased pricing was doubled in variant 3 compared to the original one.

For completeness, we also conducted an additional distribution shift study. Specifically, we asked ChatGPT to rephrase each question without changing its meaning. On the HEADLINES dataset, we observe that FrugalGPT is quite robust to such shifts: GPT-4's achieves 84% overall performance on the shifted dataset, and FrugalGPT can reach the same accuracy with 33% of the cost.

## C.4 Comparisons with output ensemble

We conducted an additional experiment for the model ensembles. In particular, we query ChatGPT 3 times with different temperatures (0, 0.1, 0.2) and then take the mode of the generation to be the final output. As shown in Table 6, sampling multiple times leads to marginal performance gains on the HEADLINES dataset. Yet, the gain is relatively smaller than FrugalGPT, whose accuracy can be 87% at the same cost.

## C.5 Effects of training dataset size

We also conducted an additional study on training set size effects on the HEADLINES dataset. Specifically, we varied the data used to train FrugalGPT from 500 to 5000, and measured the accuracy of the FrugalGPT LLM cascade on the test set with 5000 samples. As shown in Table 7, one thousand samples are sufficient to learn FrugalGPT strategies with high performance. In fact, FrugalGPT's performance was consistently better than GPT-4 (85.5%), even using only 500 samples.

# D   Prompt Details

Here we provide the prompts used for each task. In a nutshell, we use few-shot prompting for robust performance.

## D.1   Prompt for HEADLINES

Please determine the price direction (up, down, neutral, or none) in the following news headlines.

Q: december gold down $1 at $749 an ounce on nymex
A: down

Q: august gold up $7.60 at $878.80 an ounce on nymex
A: up

Q: commodity outlook: gold may find it tough to top 30,920 level
A: none

Q: gold prices at 1-week lows as dollar remains supported
A: neutral

Q: illegal flow of gold to nepal from india across porous border showing upward swing
A: none

Q: gold prices steady in early asia trade with focus on hong kong
A: neutral

Q: gold adds 0.7% to trade at record $1,601.50/oz
A: up

Q: gold loses sheen on muted demand, silver also eases
A: down

## D.2   Prompt for OVERRULING

Context: because jones/walker relates only to sufficiency of the evidence, we hereby disavow the language holding otherwise in sandoval.
Question: Is it overruling?
Answer: Yes

Context: insofar as the givens case holds contrary to our original opinion herein or to the rule expressed in the carr case and the cases there cited, it is expressly overruled.
Question: Is it overruling?
Answer: Yes

Context: the court also specifically ordered defendant to "be subject to all administrative or judicial enforcement remedies available to the plaintiff as prescribed by state and federal law in a title iv-d case[.]"
Question: Is it overruling?
Answer: No

Context: according to napa auto parts, the straws drove the vehicle "for approximately six [] weeks and [] for between 500 to 600 miles prior to the accident with no incidents."
Question: Is it overruling?
Answer: No

Context: accordingly, we answer the certified question in the affirmative, disapprove deruyter, and approve the decision of the court below.
Question: Is it overruling?
Answer: Yes

## D.3 Prompt for COQA

Context: Michigan () is a state in the Great Lakes and Midwestern regions of the United States. The state's name, Michigan, is of French origins (form of the Ojibwe word) "mishigamaa", meaning "large water" or "large lake". Michigan is the tenth most populous of the 50 United States, with the 11th most extensive total area, and the largest state by total area east of the Mississippi River. Michigan's capital is Lansing, and its largest city is Detroit. Michigan is the only state to consist of two peninsulas. The Lower Peninsula, to which the name Michigan was originally applied, is often noted to be shaped like a mitten. The Upper Peninsula (often referred to as "the U.P.") is separated from the Lower Peninsula by the Straits of Mackinac, a channel that joins Lake Huron to Lake Michigan. The two peninsulas are connected by the Mackinac Bridge. The state has the longest freshwater coastline of any political subdivision in the world, being bounded by four of the five Great Lakes, plus Lake Saint Clair. As a result, it is one of the leading U.S. states for recreational boating. Michigan also has 64,980 inland lakes and ponds. A person in the state is never more than from a natural water source or more than from a Great Lakes shoreline.
Question: what separates the two peninsulas?
Answer: the Straits of Mackinac,

Context: Harry had a very small farm. He only had one cow but dreamed about having a large farm. He once asked his father Bill, "I'd like to have that land over there. How can I get it?" His father encouraged him to go and talk to the landowner to see how they could get the land. Harry said. "But we don't have enough money." His father said, "Don't worry. Go and talk to him." Several years passed. Harry had not only the land, but also several hundred cows. He had a happy life with his wife. Later, Harry's wife, Sarah, had a dream. "I want to build the biggest farm in the world." She said. They called their friend Manuel about this task. Three days later Manuel had a plan for the whole project. Then they asked, "How much will it cost?" Manuel said they needed a lot of money. "Nobody will lend us so much money to build a farm," they thought. But the manager of the bank them and their dream. A few months later, La manuel, the biggest farm in the world, was opened.
Question: Who was the farmer?
Answer: Harry

## D.4 Prompt for AGNEWS

Please answer which category (World, Sports, Business or Sci/Tech) a provided news falls into.

Q: Five-year ban for Blackburn fan One of the two Blackburn Rovers Football Club fans charged with public disorder for racially abusing Dwight Yorke has been handed a five-year ban.
A: Sports

Q: Major software pirates caught A multimillion-euro software piracy ring has been broken following synchronized raids in Athens and London yesterday, Attica police said.
A: Sci/Tech

Q: Loews to Buy Entergy-Koch Pipeline NEW YORK (Reuters) - Conglomerate Loews Corp. <A HREF="`http://www.investor.reuters.com/FullQuote.aspx?ticker=LTR.Ntarget=/stocks/` `quickinfo/fullquote">LTR.N</A>` agreed to buy an 8,000-mile natural gas pipeline system from Entergy-Koch LP for $1.14 billion on Monday, in a bid to cash in on rising U.S. demand for natural gas.
A: Business

Q: Texas A amp;M Quarterback Finds Groove Once Again Reggie McNeal switched his jersey number in the off-season, trading No. 16 for No. 1 in a salute to a departed teammate. McNeal has become the
A: Sports

Q: UPDATE 2-Rugby-Australia edge out England in Twickenham thriller Australia showed all their famed resilience to withstand a fierce fightback by England and beat the world champions 21-19 in a thunderous World Cup final repeat on Saturday.
A: Sports

Q: Ed Hardin: Bowl situation not so Peachy CHAPEL HILL; They came down from the hills Saturday, down from the hot springs and natural bridges of Virginia, deep into the heart of ACC darkness.
A: Sports

Q: Democrat Seeks Probe of Bush Aides' Travel (AP) AP. The chairwoman of the House Democrats' homeland security task force is asking Congress' independent auditors to examine travel by senior Bush administration officials in light of recent trips to hotly contested states in the 2004 presidential election.
A: World

Q: Chelsea: #39;No fear #39; factor in Europe IF a football team could be entered in that famous television programme Fear Factor, then Jose Mourinho would register Chelsea. Because if Chelsea are going to reach the final of the Champions League for the
A: Sports

## D.5 Prompt for SCIQ

Please answer the following questions concisely.

Context: inside these cells, glucose is immediately converted into glucose-6-phosphate. By doing this, a concentration gradient is established where glucose levels are higher in the blood than in the cells. This allows for glucose to continue moving from the blood to the cells where it is needed. Insulin also stimulates the storage of glucose as glycogen in the liver and muscle cells where it can be used for later energy needs of the body. Insulin also promotes the synthesis of protein in muscle. As you will see, muscle protein can be catabolized and used as fuel in times of starvation. If energy is exerted shortly after eating, the dietary fats and sugars that were just ingested will be processed and used immediately for energy. If not, the excess glucose is stored as glycogen in the liver and muscle cells, or as fat in adipose tissue; excess dietary fat is also stored as triglycerides in adipose tissues. Figure 24.21 summarizes the metabolic processes occurring in the body during the absorptive state.

Question: Excess dietary fat is stored as triglycerides in the body. what type of tissue is used to store the triglycerides?
Answer: adipose

Context: trees that lose their leaves once a year.
Question: What are trees that lose their leaves during winter called?
Answer: deciduous

Context: Primary Vesicles As the anterior end of the neural tube starts to develop into the brain, it undergoes a couple of enlargements; the result is the production of sac-like vesicles. Similar to a child's balloon animal, the long, straight neural tube begins to take on a new shape. Three vesicles form at the first stage, which are called primary vesicles. These vesicles are given names that are based on Greek words, the main root word being enkephalon, which means "brain" (en- = "inside"; kephalon = "head"). The prefix to each generally corresponds to its position along the length of the developing nervous system. The prosencephalon (pros- = "in front") is the forward-most vesicle, and the term can be loosely translated to mean forebrain. The mesencephalon (mes- = "middle") is the next vesicle, which can be called the midbrain. The third vesicle at this stage is the rhombencephalon. The first part of this word is also the root of the word rhombus, which is a geometrical figure with four sides of equal length (a square is a rhombus with 90° angles). Whereas prosencephalon and mesencephalon translate into the English words forebrain and midbrain, there is not a word for "four-sided-figure-brain. " However, the third vesicle can be called the hindbrain. One way of thinking about how the brain is arranged is to use these three regions—forebrain, midbrain, and hindbrain—which are based on the primary vesicle stage of development (Figure 13.3a).
Question: As the anterior end of the neural tube starts to develop into the brain, it undergoes a couple of enlargements; the result is the production of these?
Answer: sac-like vesicles

Context: The immune response mainly involves the lymphatic system. The lymphatic system is a major part of the immune system. It produces leukocytes called lymphocytes. Lymphocytes are the key cells involved in the immune response. They recognize and help destroy particular pathogens in body fluids and cells. They also destroy certain cancer cells.
Question: What system of the body is most involved in the immune response?
Answer: lymphatic system

Context: Carbohydrates are organic molecules that consist of carbon, hydrogen, and oxygen. They are made up of repeating units called saccharides. They provide cells with energy, store energy, and form structural tissues.
Question: What are organic molecules that consist of carbon, hydrogen, and oxygen called?
Answer: carbohydrates

Context: Materials that are poor conductors of thermal energy are called thermal insulators. Gases such as air and materials such as plastic and wood are thermal insulators.
Question: What are materials that cannot conduct thermal energy efficiently known as?
Answer: thermal insulators

Context:
Question: In our wildflower population, the pool of what remains constant from one generation to the next?
Answer: genes

Context: The relative sizes of the atoms show several trends with regard to the structure of the periodic table. Atoms become larger going down a column and smaller going across a period.

> Question: The relative sizes of the atoms show several trends with regard to what visual method of organization?
> Answer: periodic table

## E  Additional Discussions

**Design space constraints due to black-box access.**  Note that the design space of FrugalGPT is restricted by the black-box access to LLM APIs. In this paper, we only assume access to each LLM's response to a user query, which holds for all LLM APIs considered in this paper. Relaxation of this assumption gives FrugalGPT more information and thus leads to potentially more effective design. For example, if the token likelihood value/vector is available, then one may use it to quantify an LLM's own uncertainty about its generation for a more effective scorer. Alternatively, information on some LLM services' expertise (e.g., through its training data) may also be helpful.

**Applications to scenarios requiring transparency and model control.**  This paper mainly focuses on using black-box APIs such as GPT-4. This is because these APIs have been widely used in many real-world tasks, due to their easiness to use and excellent performance on many tasks. With that being said, FrugalGPT is also applicable to scenarios requiring transparency and control, where local models with white-box access are needed. In these scenarios, users can simply use FrugalGPT with open-source models, providing access to these models' responses and cost estimation.

**Evaluation scopes.**  Tasks with short outputs often come with ground-truth labels that enable objective evaluations. Many LLM benchmarks and evaluations rely on short outputs or multiple-choice questions. Tasks with short outputs also appear in many real-world applications. Quality evaluation for long generations, on the other hand, often involves LLM as a judge, with potential biases and subjectiveness. Thus, we focus on the former to ensure the evaluation is objective and matches human judgments. Furthermore, this paper mainly focuses on improving performance and reducing cost for relatively short queries. We leave a detailed study on long-form queries as future work.

**Challenges for training the scorer.**  There is no numerical label for quality, and thus we cannot use the standard regression training paradigm directly. Instead, we trained a model to predict correctness, which is binary and can be curated from the ground-truth labels. Then we used the model's confidence as the judge's quality estimation. Empirically, we observe that the model's confidence is highly correlated to true quality.

**Justification for the chosen datasets.**  We chose the datasets because they represented an important subset of real-world applications. For example, the HEADLINES and OVERRULING datasets represent the task of text classification and tagging, which is listed as an important application of GPT-4 by OpenAI (see, e.g., `https://platform.openai.com/examples/default-tweet-classifierandhttps://platform.openai.com/examples/default-review-classifier`). The SCIQ dataset measures the scientific question-answering ability, which is key for QA applications such as the Khan academy pilot program (see `https://openai.com/gpt-4`). Furthermore, they cover a diverse set of tasks. For example, COQA and SCIQ are free-form QA datasets, and HEADLINES, OVERRULING, AGNEWS are classification datasets.

**More implementation details and potential overfitting concerns.**  To ensure fair comparisons, we use the same temperature, maximum tokens, and the stop token for all models evaluated in this paper. Since all tasks are objective, we set the temperature at 0.1 to discourage diverse generations. As the desired answers across our tasks are short, we set the maximum tokens at 50. The stop token used in this paper is the enter key "\n". Overfitting is a common concern in ML practice. As shown in Table 7, FrugalGPT's performance remains stable as long as the training data size is at least one thousand. This suggests that FrugalGPT is not likely to overfit as long as the datasets are not too small.