# OpenReview forum: "FrugalGPT: How to Use Large Language Models While Reducing Cost and Improving Performance"
_TMLR — Accepted by TMLR_

### Review · Reviewer_knVh · 2024-07-28

**Summary Of Contributions:**

The paper introduces a method which jointly trains a routing model and a scorer to (1) choose which of several LLMs to send a query to and (2) decide whether the result is of sufficient quality. The method results in significant time savings on 2 to 4-way classification datasets (>50x on one likely easy dataset and 4x on two others), and ~2x speedups up two question answer datasets.

**Audience:**

No

**Broader Impact Concerns:**

No broader concerns: I agree with the authors' description of ethical aspects based on efficiency gains.

**Claims And Evidence:**

Yes

**Requested Changes:**

The key two changes are as mentioned above:

1. Compare against the baseline of using a single scalar threshold on GPT-J logits to decide whether to accept the result of one GPT-J call or call up to GPT-4. This is a simplified version of L = 2 that avoids a training step. For classification and multiple choice tasks this is immediate; for the free-text QA task it may be useful to divide by the number of tokens.
2. Highlight the types of datasets in Table 2, with a column that says "<n>-way classification", "multiple-choice QA", or "free-text QA" (or equivalent, and discuss the differences in results between the different types.

Here are some smaller requested changes:

1. CoHere shouldn’t have a capital H.
2. The “Ope (b)” citation looks like a typo. There are a few others like this.
3. Add a limitation about long-form queries, as I would expect this method to produce smaller speedups there.
4. The 5% quality improvement on some datasets seems likely to be a real ensembling effect, but it should be presented as such (that it likely has the same cause as ensembling methods).
5. Figure 5 in the appendix says the details can be found in the appendix.

**Strengths And Weaknesses:**

The main strength of the paper is that the general approach of calling smaller models before bigger models is solid, and the results seem solid as well besides the dataset type issue below.

The main critique of the paper is that it is quite elaborate, and there is no comparison to a far simpler available baseline: send the answer to GPT-J and accept sufficiently high probability answers (this is a simplified version of their L = 2 setup, but requiring training only a single scalar for the classification case). There is a brief allusion to this strategy in the first paragraph of Appendix E, implying they intentionally avoided such strategies because some LLM APIs do not provide log probs. However, returning log probs does not add any cost to an API, and the feature is missing from proprietary model services for security reasons alone (anti-model theft and jailbreaking). Smaller open models such as GPT-J do provide log probs: https://textsynth.com/documentation.html#logprob

Another concern is that there is no discussion of differences between the datasets that might be contributing the quality variation. Citing a clear outlier in performance in the abstract is bad: any sufficiently easy classification dataset would get such a result (which the simpler baseline from the previous paragraph), the other classification datasets get only 4x, and the QA datasets can only 2x. This distinction is hard to see from the main text of the paper: it occurs only in prose in an appendix. The type of dataset (n-way classification vs. multiple choice QA vs. free-text QA) should be given in the main body, ideally in Table 2.

Smaller notes are listed below.

---

### Review · Reviewer_YQvh · 2024-08-18

**Summary Of Contributions:**

This paper introduces FrugalGPT, a framework designed to optimize the utilization of Large Language Models (LLMs) by dynamically selecting the most suitable LLM for specific tasks and queries, thereby minimizing costs.
Frugal GPT learns a generation judger to assess the quality of LLM outputs and then sequentially invoking LLMs until the judger's score meets a predefined threshold.
They empirically show that FrugalGPT can save up to 98% of the inference cost of the best individual LLM API while matching its performance on the downstream task. Additionally, FrugalGPT can improve performance by up to 4% at the same cost.

**Audience:**

Yes

**Broader Impact Concerns:**

None.

**Claims And Evidence:**

Yes

**Requested Changes:**

- The prices of LLM APIs for Inference has gone down significantly compared to the prices reported in Table 1. Including the cost savings considering the new prices will strengthen this paper.
- Adding more information on how the LLM router works in the Frugal GPT pipeline will strengthen the paper.
- Further elaboration on implementation details, theoretical analysis, and potential overfitting concerns would enhance the paper’s technical depth and contribution.

**Strengths And Weaknesses:**

Strengths:
- The paper addresses a critical challenge in widespread adoption of LLMs: the high cost of inference and offers a very simple and practical solution: intelligently routing queries to different LLMs based on their cost and performance characteristics.
- The paper is generally well-written and organized, making it easy to follow the authors' ideas and contributions.

Weaknesses:
- The framework relies on the black-box nature of LLM APIs, which may limit its applicability in scenarios where greater transparency and control over the models are required.
- The paper considers models that are slightly out of date. Models such as GPT-4o, GPT-4o mini and Gemma-2B aren’t included.
- The paper assumes the availability of a reliable answer scorer that can accurately assess the quality of LLM-generated answers. However, the quality of such a scorer can significantly impact the overall performance of FrugalGPT. The paper briefly mentions an ablation study on the generation scorer in Section 5, but a more detailed discussion of the scorer's limitations and potential impact on FrugalGPT would be valuable.

---

### Review · Reviewer_ymTi · 2024-09-05

**Summary Of Contributions:**

To save cost and improve accuracy, this paper proposes FrugalGPT, an algorithmic framework to adaptively select the generative LLM APIs to sue for different queries. FrugalGPT has 3 components: a LLM router, an answer-scorer, and a stop judger. The experiments demonstrate that the proposed framework can match the performance of the best individual generative LLM or betterwith up to a 98% cost reduction.

**Audience:**

Yes

**Broader Impact Concerns:**

I did not see any broader impact concerns.

**Claims And Evidence:**

Yes

**Requested Changes:**

See the above weaknesses. While the proposed framework demonstrates superior performance on several QA and classification tasks, there are several concerns: 1. LLMs are better at generative tasks other than classification tasks. The main usage of LLMs is focused on generative tasks; 2. Since there are consistent prices drops of state-of-the-art LLMs and the gap between open-source LLMs (e.g., Llama3.1) is smaller, is it necessary to use the proposed framework?

**Strengths And Weaknesses:**

**Strengths**

1. The proposed framework is simple but effective.
2. There is a significant cost reduction while using reasonable scorer models to route the queries.


**Weaknesses**
1. In page 4, the input and response costs of GPT-4 are $30 and $60 per million tokens. The calculation is wrong which should be $2.12K instead. Also this price does not match the numbers in Table 1.
2. While the chose datasets represent a subset of real-world applications, there is not enough justification for these classification datasets. Some smaller models, such as roberta, should be able to match the performance of many LLMs. In general, LLMs are better at generative tasks, such as open domain QAs, which lacks experimental analysis.
3. It is not clear whether the proposed framework is task-specific or task-agnostic. Does the hyperparameters need to be tuned for each task?
4. It seems that the proposed framework is highly dependent on the behavior of several LLM APIs. This may make the proposed framework less stable when applied to real-world applications.

---

### Comment · Editors_In_Chief · 2025-11-28

Hi authors, I just wanted to inform you that, after revisiting some previously accepted papers, we've decided to grant this paper a Featured Certification. Congratulations!

Gautam, on behalf of the Editors in Chief

---

### Decision · Action_Editor_xAXZ · 2024-11-04

**Recommendation:** Accept with minor revision

**Comment:**

Following the authors' revision, two reviewers lean toward acceptance, acknowledging the positive contributions of this paper. One reviewer (Reviewer ymTi), however, leans toward rejection, with remaining concerns including: (1) the sustainability of the proposed framework given the rise of proprietary LLM APIs with substantially reduced costs and the availability of open-source alternatives; (2) the need for a more comprehensive evaluation on generative tasks; and (3) the complexity of the proposed framework.

I agree with Reviewer ymTi that these concerns are valid. However, I believe that the contributions of this work offer value to the community despite these limitations. In the revision, the authors included a comparison with a simpler baseline, which helps validate the necessity of the proposed approach’s complexity. The authors also demonstrated the effectiveness of their approach with more recent LLM models and on two generative tasks.

​​Given these points, I recommend acceptance of this paper. I have the following suggestions for the authors as they prepare the final version:

* The newly added comparisons with a simpler baseline (C.6) and more recent LLM models (C.7) are essential to demonstrate the necessity of the approach’s complexity and its current effectiveness. Including these results in the main paper would strengthen the validation of the method.

* If possible, adding further results on generative tasks could enhance the paper, as suggested by Reviewer ymTi.

**Audience:**

Researchers and practitioners focused on LLMs and their real-world applications – especially those interested in balancing performance and cost – may find this paper valuable. Additionally, researchers interested in predicting and evaluating model performance may discover some of the paper's findings to be particularly relevant.

**Claims And Evidence:**

Summary:

This paper proposes FrugalGPT, an effective approach that dynamically selects the most appropriate LLM API for each query to minimize costs while preserving performance. The key idea involves using an LLM judger to evaluate the quality of LLM outputs and then sequentially invoking LLMs until the judger’s score reaches a predefined threshold. Evaluations on widely-used benchmarks demonstrate the effectiveness of FrugalGPT, achieving performance comparable to the best individual LLM while saving 98% on inference costs.

Claims:

The key claims made in the paper are that (1) for many tasks that utilize LLMs, it is possible to assess the result quality using small models; (2) no single LLM is universally superior to the others; (3) the proposed FrugalGPT adaptively routes queries to different LLMs, effectively reducing costs while enhancing performance; and (4) FrugalGPT is effective across multiple downstream tasks, such as news classification, reading comprehension, and scientific question answering, and demonstrates compatibility with real-world LLMs, including ChatGPT, GPT-3, GPT-4, and J1.

Evidence:

The claims are well supported by the characteristics of the proposed method and by the experimental results presented in both the original submission and the revision.

---

> ### Author Response · Authors · 2024-12-03
> **Thank you for your acceptance and suggestions!**
>
> Thank you very much for your acceptance and suggestions! We have revised the paper accordingly and uploaded the updated paper. In particular, we have evaluated FrugalGPT's performance on several datasets using 10 recent models including Claude 3.5 Sonnet, GPT-4 Turbo, GPT-4o, and Google Gemini 1.5 Pro, and highlighted the results in the main paper (Figure 1, Table 3, and Figure 4, and Figure 5). The abstract and introduction have also been updated to reflect the more recent results.
>
> Thank you again for your acceptance and constructive suggestions!